# Increased spatial extent and likelihood of compound long-duration dry and hot events in China, 1961-2014

Yi Yang[1,2], Douglas Maraun[2], Albert Ossó[2], Jianping Tang[1]

[1]School of Atmospheric Sciences, Nanjing University, Nanjing, 210023, China
5   [2]Wegener Center for Climate and Global Change, University of Graz, Brandhofgasse 5, 8010, Graz, Austria

*Correspondence to*: Douglas Maraun (douglas.maraun@uni-graz.at), Jianping Tang (jptang@nju.edu.cn)

**Abstract.** Compound dry and hot events can cause aggregated damage compared with isolated hazards. Although increasing attention has been paid to compound dry and hot events, the persistence of such hazard is rarely investigated. Moreover, little attention has been paid to the simultaneous evolution process of such hazard in space and time. Based on observations during 10  1961-2014, the spatiotemporal characteristics of compound long-duration dry and hot (LDDH) events in China during the summer season are investigated on both a grid basis and a 3D event basis. Grid-scale LDDH events mainly occur in eastern China, especially over northeastern areas. Most regions have experienced a pronounced increase in the likelihood of LDDH events, which is dominated by increasing temperatures. From a 3D perspective, 146 spatiotemporal LDDH (SLDDH) events are detected and grouped into nine spatial patterns. Over time, there is a significant increase in the frequency and spatial 15  extent of SLDDH. Consistent with the grid-scale LDDH events, hotspots of SLDDH events mainly occur in northern China, such as Northeast, North China and Qinghai clusters, which are accompanied by high occurrence frequency and large affected areas greater than 300,000 km$^2$.

## 1 Introduction

Extreme weather and climate events such as droughts and heatwaves are commonly analyzed in terms of univariate statistics 20  like dry spells and heatwave intensity (Dosio et al., 2018; Trenberth et al., 2014). However, extreme events often result from a combination of interacting drivers and hazards. Such compound events can lead to more severe environmental and societal impacts than single hazards (Leonard et al., 2014; Zscheischler et al., 2020). Therefore, traditional risk assessment considering single drivers may substantially underestimate the associated risks (Raymond et al., 2020). A typical compound event in almost all regions of the world is compound dry and hot (DH) events, which have devastating effects on agricultural 25  production, water security, and human health (Pfleiderer et al., 2019; Zscheischler and Fischer, 2020).

Compound dry and hot events have received increasing attention in recent years. Both observations and model simulations suggest more frequent compound hot and dry conditions in recent decades across the entire globe (Hao et al., 2018; Sarhadi et al., 2018). At the regional scale, a substantial increase in concurrent dry and hot extremes has been observed in the United States (Alizadeh et al., 2020; Mazdiyasni and AghaKouchak, 2015), India (Sharma and Mujumdar, 2017), Europe (Manning

et al., 2019; Sutanto et al., 2020), and China (Chen et al., 2019; Li et al., 2019; Lu et al., 2018). There is strong evidence that future global warming will lead to a higher likelihood of compound dry and hot summers (Coffel et al., 2019; Sarhadi et al., 2018; Zscheischler and Seneviratne, 2017).

Most previous studies focus on the occurrence frequency and severity of compound dry and hot events. However, the persistence of compound extremes is rarely investigated. Prolonged periods of dry and hot conditions, such as those over
Europe in 2018, can increase the risks of soil moisture droughts and wildfires and cause tremendous adverse impacts on agriculture and society (Manning et al., 2018). Given the severe impact of persistent extremes, effective risk management requires a comprehensive assessment of the characteristics and possible changes in such long-duration extreme events. Previous studies indicate that summer weather has become more persistent over recent decades due to anomalous atmospheric circulation patterns and global warming (Coumou and Luca, 2020; Pfleiderer et al., 2019). More long-duration
extremes have been observed recently (Breinl et al., 2020; Kornhuber et al., 2019; Zscheischler and Fischer, 2020). It is not yet clear whether the characteristics of persistent compound dry and hot events have changed in the past decades and what the potential drivers are. Furthermore, most studies identify compound dry and hot conditions using accumulated precipitation deficits at monthly or seasonal time scales, for instance, with the standardized precipitation index (SPI, Mckee et al., 1993), which does not quantify the duration of the events. While recent work by Yu and Zhai (2020) highlighted an
increased persistence of compound dry and hot events averaged across China, the regional patterns and the potential drivers of changes for the occurrence of compound long-duration dry and hot (LDDH) events in China remain unexplored.

Moreover, existing studies mainly identify DH events at individual stations or grid points, based on which they estimate the regional characteristics of DH events. But extreme events such as droughts and heatwaves are not local (grid-point) phenomena and generally have a specific impacted area and duration (Ren et al., 2012; 2018). There is a lack of systematic
research on the spatial characteristics (3D structure) of compound dry and hot events. An event-based identification of spatiotemporal LDDH (SLDDH) could facilitate tracking the daily spatiotemporal dynamics of SLDDH and understanding the associated physical drivers. The detection of SLDDH needs data with sufficient resolution that provide large-scale climate information, such as satellite measurements and large weather station networks. Further, while few studies have found a significant increase in the spatial extent of compound dry and hot events in different regions of the world (Sharma
and Mujumdar, 2017; Yu and Zhai, 2020), they focus on DH events at specific stations or grid points and consider the spatial extent as the fraction of land covered (not necessarily contiguously). The spatial extent, specifically, contiguous areas simultaneously affected by a DH event (Vogel et al., 2020), is rarely examined.

Here we analyze the spatial and temporal variations of the compound long-duration dry and hot events over China on both a grid basis and a 3D event basis. Following Manning et al. (2019), we define compound long-duration dry and hot events at a
grid scale as extended dry periods that co-occur with extreme temperatures. This definition using a daily time step allows us to better understand the features of individual events. Spatiotemporal compound long-duration dry and hot events are detected and quantified, considering both the spatial and temporal coherence of concurrent dry and hot processes.

## 2 Data and Methods

### 2.1 Data

The daily precipitation and daily maximum temperature (Tmax) from the homogenized gridded observational dataset CN05.1 are used to identify and characterize LDDH events from 1961 to 2014 (Wu and Gao, 2013). This dataset is based on interpolations from 2416 meteorological stations across China and has a spatial resolution of 0.25°. It has been widely applied for model evaluation and climate change detection (Yang et al., 2019b; Zhou et al., 2016). Note that the station density in western China is lower than in eastern China, leading to a great uncertainty in this region, with the largest

uncertainty over the northern part of the Tibetan Plateau and Taklimakan desert in southern Xinjiang (Peng and Zhou, 2017; Wu and Gao, 2013).

### 2.2 Event definition

#### 2.2.1 Grid-scale compound long-duration dry and hot events

We identify LDDH events locally following the definition by Manning et al. (2019). Long-duration dry and hot events are

persistent dry spells accompanied by extremely hot temperatures and can be characterized by their duration (DUR) and magnitude (MAG). Here DUR is the number of consecutive days with precipitation below 1 mm, and MAG of each event is the hottest daily maximum temperature during the dry period. Thus, for each dry spell, there is a (DUR, MAG) pair. DUR is similar to the definition of CDD (consecutive dry days, Sillmann et al., 2013), while MAG is similar to TXx (maximum value of daily maximum temperature, Alexander et al., 2006) and HWA (heatwave amplitude, Perkins and Alexander, 2013).

To account for regional differences, in each grid cell, the concurrent exceedances of DUR and MAG above the individual 95th percentiles (i.e., $DUR_{95}$ and $MAG_{95}$) during 1961-2014 are defined as compound long-duration dry and hot events.

Our definition of persistent meteorological droughts using a percentile threshold of the number of consecutive dry days is consistent with previous studies. For example, using the 80th percentile of dry spell duration, Raymond et al. (2016) revealed the spatiotemporal characteristics of prolonged dry spells in the Mediterranean and analyzed the associated synoptic

atmospheric conditions (Raymond et al., 2018). Zhang et al. (2019) gave a definition based on the 95th percentile of CDD for the extreme drought in northern China. Building on previous work on the persistent meteorological droughts, we introduce a high temperature threshold ($MAG_{95}$) to identify compound long-duration dry and hot conditions. Our results are not sensitive to the choice of climatological periods and percentile thresholds used to define the LDDH events (not shown). We focus our analysis on LDDH events in boreal summer (June, July and August) because of their high frequency and

severe socioeconomic effects during this season (Ridder et al., 2020).

Once grid-scale LDDH events are identified, five metrics are calculated for each year and each grid-box. These include:

- Count: the number of LDDH events occurring in a year;
- Duration: the annual average DUR across events in each year;
- Magnitude: annual average MAG across events in each year;

  •      Total days: the total number of LDDH days within a year, and

•      Timing: the annual average onset date across events in each year.

### 2.2.2 Spatiotemporal compound long-duration dry and hot events

When considering extreme events as a spatiotemporal phenomenon covering certain spatial areas over a period, a spatiotemporal compound long-duration dry and hot event is defined as a contiguous dry and hot region that lasts for several consecutive days. This approach allows for tracking the development of the spatiotemporal LDDH events. First, the LDDH events are identified for each grid point, and then we consider the spatial coherence and temporal persistence of these events to identify SLDDH. This algorithm is similar to the detection approach of regional drought proposed by Andreadis et al. (2005) and regional heatwave provided by Stefanon et al. (2012) and Lyon et al. (2019). The detailed detection approach is as follows (Fig. 1):

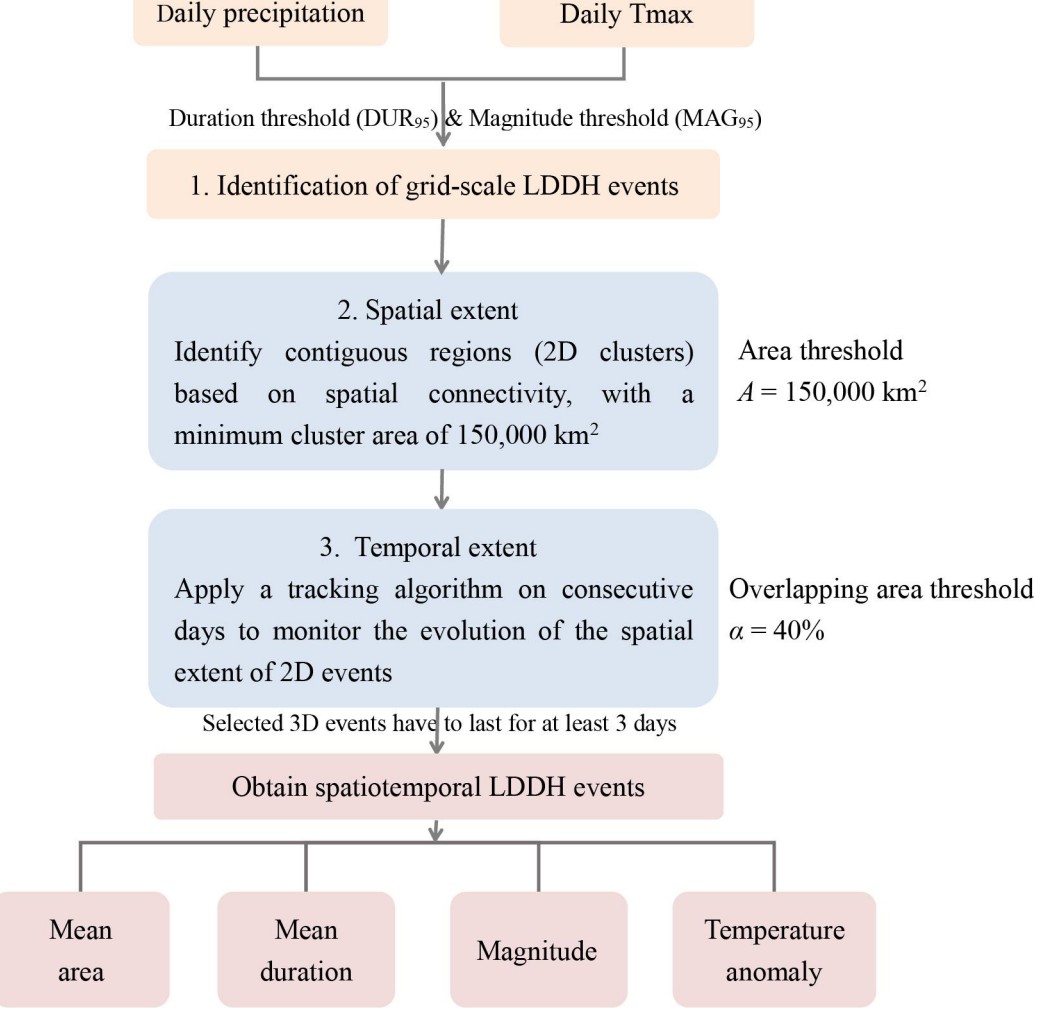

**Figure 1: Flow chart for the identification of spatiotemporal compound long-duration dry and hot events.**

1. Hazard thresholds: The grid-scale LDDH events are detected according to Sect. 2.2.1 as the joint exceedances of the thresholds $DUR_{95}$ and $MAG_{95}$.

2. Spatial extent: For each day, spatially contiguous dry and hot regions (2D clusters) are identified by applying the connected components algorithm in R package wvtool (Sugiyama and Kobayashi, 2016). A minimum area threshold A of 150,000 km$^2$ is used to eliminate the influence of small-scale extremes (Andreadis et al., 2005; Sheffield et al., 2009). This area threshold is determined through sensitivity analysis (A = 50,000, 100,000, 150,000, 200,000 km$^2$) and in line with drought studies in China based on soil moisture (Wang et al., 2011) and PDSI (Palmer Drought Severity Index, Shao et al., 2018).

3. Temporal extent: The contiguous regions under concurrent dry and hot conditions are tracked through time by looking at overlapping areas between 2D clusters on consecutive days. If the overlap is greater than 40%, the clusters are considered one single event. We have tested different overlapping area thresholds α ranging from 35% to 45%, and found little difference among different choices of α. As in Lopez et al. (2018), a minimum of three consecutive days of SLDDH events is required to exclude short-duration events.

Characteristics such as mean impacted area, mean duration, magnitude, and temperature anomaly are quantified for each spatiotemporal LDDH event. The mean impacted area is defined as the average daily spatial extent throughout the lifespan of an event. Mean duration is computed as the duration averaged across all grid points contributing to the SLDDH. The magnitude and temperature anomaly of the SLDDH events are calculated as the average daily extreme high temperature and average daily high temperature intensity, respectively. The extreme high temperature is the Tmax averaged over all grid points of the event, and the daily high temperature intensity is the average of Tmax exceeding the 1961-2014 daily climatology across all grid points within the event, both of which are calculated for each day of the SLDDH events.

### 2.3 Cluster analysis

Cluster analysis is widely used to classify extreme events (Aladaileh et al., 2019; Lopez et al., 2018; Stefanon et al., 2012; Wang et al., 2018). Here we apply a hierarchical clustering algorithm (Rokach and Maimon, 2005) to identify clusters of SLDDH events over China following their geographical location and analyze their associated characteristics. The clustering is based on Euclidean distance with the Ward criterion, which minimizes the intra-cluster variance (Raymond et al., 2016; Ward, 1963). The final number of clusters is determined according to the intra-cluster distance and Silhouette Coefficient index (Kaufman and Rousseeuw, 2009).

### 2.4 Return periods

Return period (RP) is an important way to quantify the risk of extreme events (Salas and Obeysekera, 2014). For compound events, a multidimensional RP is required. Here the bivariate return period of LDDH events is calculated using the AND case, representing the average waiting time between events where both duration and magnitude exceed the respective thresholds (i.e., $DUR_{95}$ and $MAG_{95}$). A parametric copula-based bivariate probability distribution developed by Bevacqua et

al. (2019) is applied to selected events with DUR and MAG that are simultaneously high, using the respective thresholds $DUR_{sel}$ and $MAG_{sel}$ (90th percentiles). As in Manning et al. (2019), locations with few sample size (less than 20 events) are not analyzed. These cases are found in some parts of the Tibetan plateau. The bivariate return period is calculated as:

$$
\begin{aligned}
RP(DUR_{95}, MAG_{95}) &= \frac{\mu}{P((DUR > DUR_{95} \text{ and } MAG > MAG_{95}) \mid (DUR > DUR_{sel} \text{ and } MAG > MAG_{sel}))} \\
&= \frac{\mu}{1 - F_{DUR}(DUR_{95}) - F_{MAG}(MAG_{95}) + F_{DM}(DUR_{95}, MAG_{95})} \\
&= \frac{\mu}{1 - u_{DUR95} - u_{MAG95} + C(u_{DUR95}, u_{MAG95})}
\end{aligned}
\tag{1}
$$

where $\mu$ is the average inter-arrival time between the selected events. $F_{DUR}$ and $F_{MAG}$ are the marginal cumulative distribution functions (CDFs) of DUR and MAG exceeding the selection threshold, respectively, while $F_{DM}$ is the joint probability distribution of DUR and MAG. $C$ is the copula (Serinaldi, 2015) describing the dependence between the selected (DUR, MAG) pairs.

At each grid point, we select an appropriate copula to model the dependence structure based on the Akaike information criterion (Akaike, 1974) with the R package VineCopula (Schepsmeier et al., 2016). Ten possible copula families are considered: Gaussian, t, Clayton, Gumbel, Frank, Joe, BB1, BB6, BB7, and BB8. The marginal distributions of duration and magnitude above the selection thresholds are modelled with an exponential distribution and Generalized Pareto Distribution, respectively. The goodness of fit of copulas and marginals is tested according to the Cramer-von Mises statistic (Genest et al., 2009).

## 2.5 Partitioning of return period variations

Using the method proposed by Bevacqua et al. (2019), three experiments are used to assess the contribution of changes in event (1) duration, (2) magnitude, and (3) their dependence to the changes in the return period. This method has been successfully applied to attribute changes in DH events (Manning et al., 2019) and compound flooding (Bevacqua et al., 2020) to their underlying drivers. As in previous studies (Kong et al., 2020; Manning et al., 2019; Sharma and Mujumdar, 2017), available records are divided into two halves to quantify the changes in past decades: 1961-1987 (*ref*, reference period) and 1988-2014 (*pres*, present period). For experiment *i*, the relative change in the return period is given as

$$
\Delta RP_{exp\ i} = \frac{RP_{exp\ i}^{pres} - RP^{ref}}{RP^{ref}} \times 100
\tag{2}
$$

where $RP^{ref}$ and $RP_{exp\ i}^{pres}$ are the return periods for the reference period and each experiment, respectively. Experiment 1 only considers the variation in the overall marginal distribution of duration from the reference to the present period. Given the variables $DUR^{ref}$ and $DUR^{pres}$, we first obtain $DUR_{exp\ 1}$, which has the marginal distribution of DUR in the present period using $DUR_{exp\ 1} = F_{DUR^{pres}}^{-1}(F_{DUR^{ref}}(DUR^{ref}))$, where $F_{DUR^{ref}}$ and $F_{DUR^{pres}}$ are the empirical CDFs. Then we compute the

return period $RP_{exp\,1}^{pres}$ based on $(DUR_{exp\,1}, MAG^{ref})$ with Eq. (1). The effect of changes in magnitude distribution (experiment 2) can be calculated similarly. In experiment 3, we define $DUR_{exp\,3} = F_{DUR^{ref}}^{-1}(F_{DUR^{pres}}(DUR^{pres}))$ and $MAG_{exp\,3} = F_{MAG^{ref}}^{-1}(F_{MAG^{pres}}(MAG^{pres}))$. Thus $DUR_{exp\,3}$ and $MAG_{exp\,3}$ share the same Spearman correlation and tail dependence with that of the present, but the marginal distribution is that during the reference period. The joint probability using $DUR_{exp\,3}$ and $MAG_{exp\,3}$ allows us to illustrate how the changes in the dependence structure influence the likelihood variations. The significance of the return period changes is identified by comparing $RP_{exp\,i}^{pres}$ with the 90% confidence interval of $RP^{ref}$ due to natural variability. This confidence interval is estimated with a bootstrap approach (Bevacqua et al., 2020; Guerreiro et al., 2018; Manning et al., 2019) by resampling the calendar years of precipitation and Tmax bivariate time series 1,000 times, which could preserve the autocorrelation of the variables.

## 3 Results and Discussion

### 3.1 Grid-scale compound long-duration dry and hot events

#### 3.1.1 Spatial and temporal characteristics

Figure 2 shows the spatial pattern of climatological summertime frequency, duration, magnitude, total days and onset timing of grid-scale LDDH events during 1961-2014. Large spatial variations are prevalent in the frequency of LDDHs, with higher occurrences in eastern than in western China. Grid-scale LDDH events occur most frequently over Northeastern China, and to a lesser extent, along the middle and lower reaches of the Yangtze River Basin (YRB). The duration of LDDH events increases from southeast to northwest of China (Fig. 2b). Although with a relatively low occurrence frequency, LDDH events in Northwest China are most persistent, with a duration of more than 60 days. Durations are shorter at the southeast coast (around 7 days). Regarding the temperature magnitude of the LDDH events (Fig. 2c), higher values are located in Northwest China, followed by central-eastern China, whereas the Tibetan Plateau with high altitudes has lower temperatures. The total number of LDDH days (Fig. 2d) is heterogeneously distributed across the country, with the northern regions and YRB experiencing relatively more concurrent dry and hot days.

There are pronounced differences in the onset timing of LDDH events in summer (Fig. 2e). In Yunnan and North China, grid-scale LDDH events, on average, start in the early and mid-June. Relatively late occurrence of LDDH events is found over the Yangtze River Basin, where the concurrent extremes generally start in mid-to-late July. This is consistent with the northward advance of the East Asian summer monsoon (Ding and Chan, 2005). With the rain belt located in northern China, the Yangtze River Basin is under the control of the western Pacific subtropical high, which is conducive to the formation and development of extreme high temperature and drought (Li et al., 2015; Wang et al., 2016). The results compare well with those based on gauge observations (not shown), confirming the validity of the gridded dataset for this analysis.

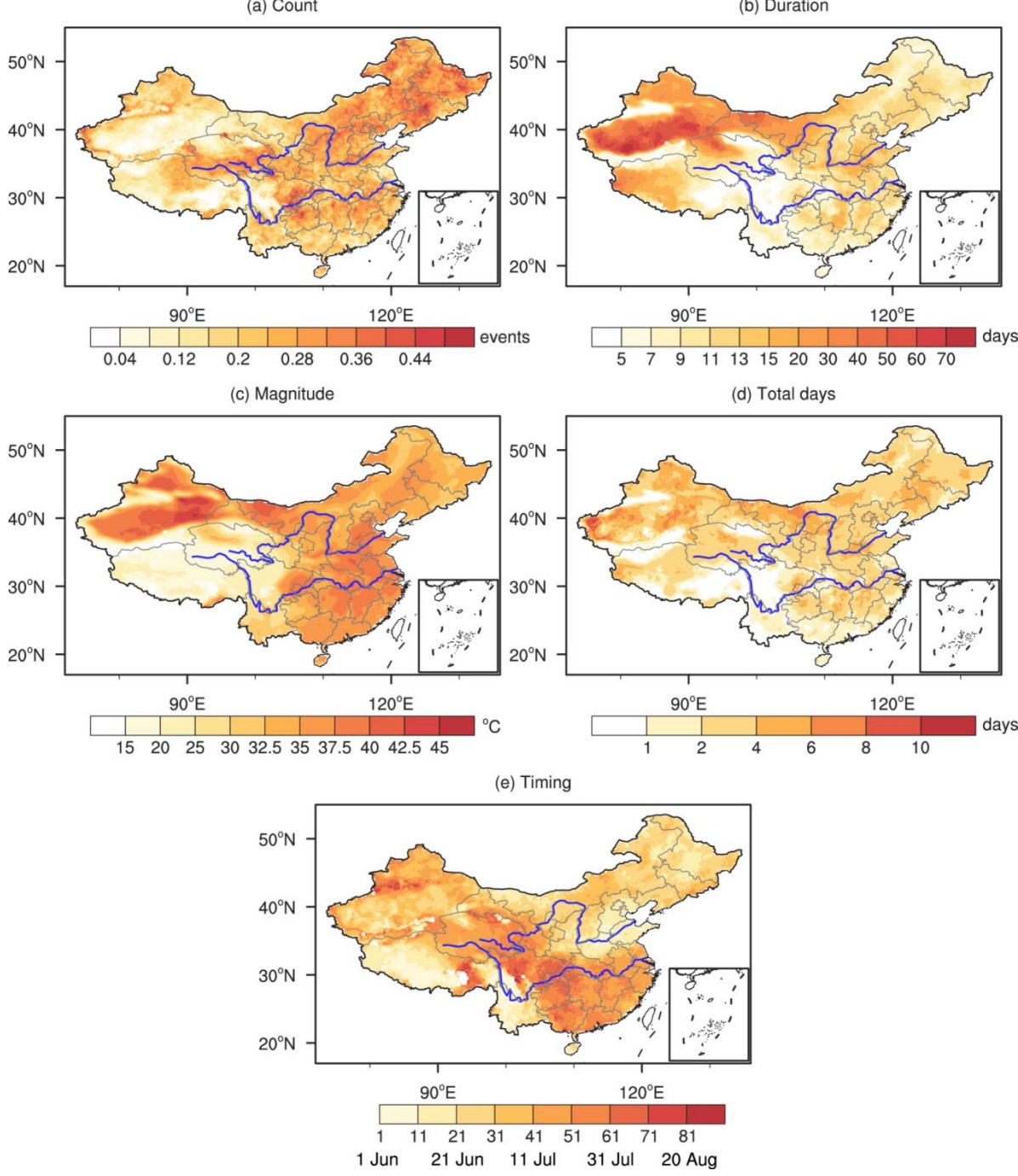

Figure 2: Maps of grid-scale compound long-duration dry and hot events characteristics in China. Shown are the annual mean (a) number of events, (b) duration, (c) magnitude, (d) total number of days, and (e) onset timing over the period 1961-2014. The timing is represented as the days since June 1, and 1 corresponds to June 1.

The spatial distribution of linear trends of grid-scale LDDH event number and total days from 1961 to 2014 are presented in Fig. 3. Most parts of China have experienced an increased occurrence of compound long-duration dry and hot events (Fig.

3a). The largest increase has occurred in Northeast China, Inner Mongolia, Yunnan in Southwest China and the coastal area of southeastern China. Conversely, the grid-scale LDDH frequency decreases over western Northwest China and central-eastern China. This agrees with Zhang and Zhou (2015), who have found a decrease in drought conditions over Northwest China since the 1950s. Similar results have been reported by Li et al. (2019), which identified DH events in northwest China based on a monthly drought index using station data. The trend distribution of LDDH days (Fig. 3b) largely mirrors that of

the number of LDDH events, with an increasing trend in most areas. Averaged across the whole country, both the number of events and LDDH days exhibit a statistically significant increasing trend over China (Fig. 3c-d), indicating that persistent DH events have become more frequent.

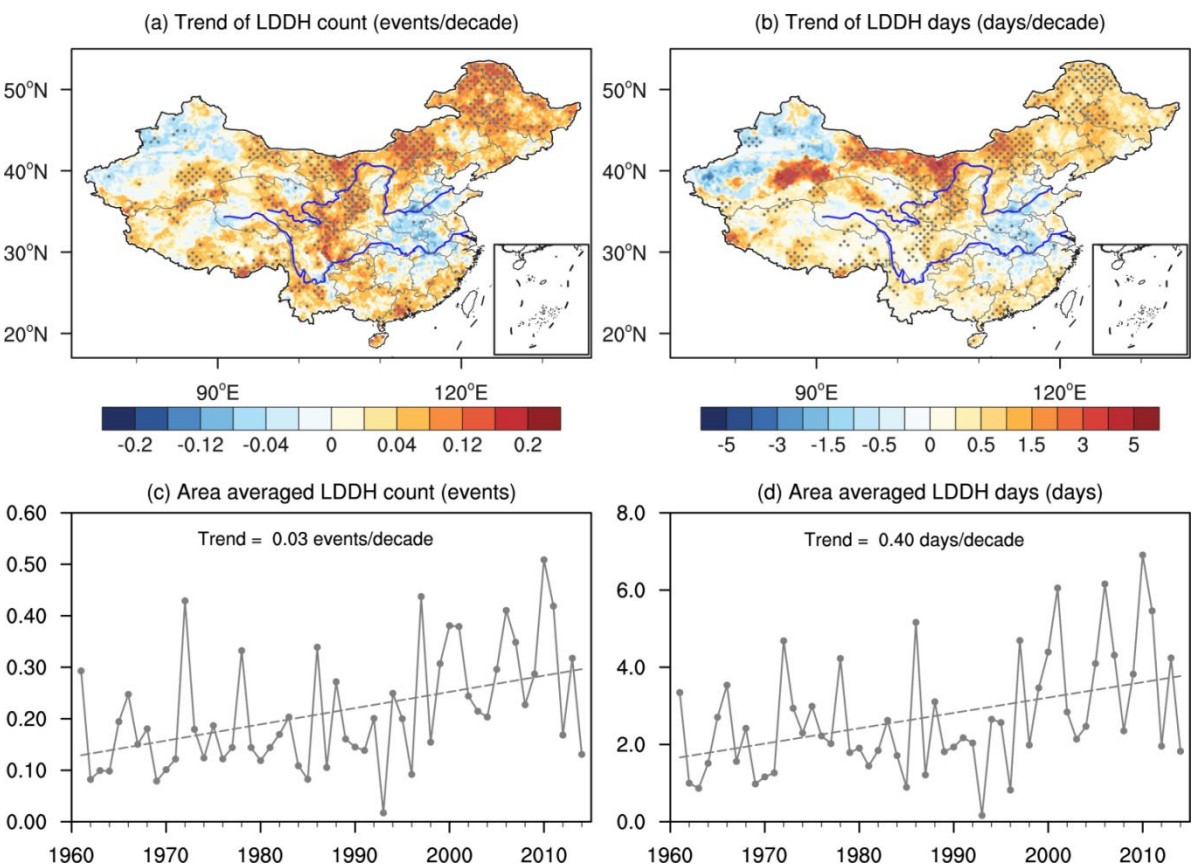

**Figure 3: Trends in the seasonal number of grid-scale (a) LDDH events and (b) LDDH days over the period 1961-2014. Stippling denotes statistically significant trends at the 90% confidence level. Regionally averaged time series of annual mean LDDH (c) count and (d) total days. The dashed line shows the linear trend.**

### 3.1.2 Changes in the LDDH probability and drivers of changes

In this section, we investigate the occurrence probability of grid-scale LDDH events using a multivariate framework following Manning et al. (2019). We divide the observations into two halves to assess whether compound extremes have changed in the past decades. The geographical occurrence probability of grid-scale LDDH events expressed as the return period is shown in Fig. 4. Regions with low return periods are more likely to suffer from this hazard. A robust difference is found for the LDDH events between the two periods (Fig. 4a-b). During 1961-1987, prolonged meteorological drought and extremely high temperatures frequently co-occur in central-eastern China, Northeast China, and western Northwest China. During the present period (1988-2014), return periods decrease by more than 40% relative to 1961-1987 across large areas of China, especially over Northeast China, Inner Mongolia, Sichuan Basin, southeast coast and some areas in the Tibetan plateau (Fig. 4c), corresponding to an increase in occurrence probability by more than 67%. Areas with a pronounced increase in LDDH probability between the two periods generally coincide with regions exhibiting a significant positive trend in the number of grid-scale LDDH events during the entire observation period (Fig. 3a). This indicates the robustness of changes in grid-scale LDDH events. Compared to 1961-1987, the fraction of areas experiencing return periods lower than 5 years increases from 33% to 59% in 1988-2014. In contrast, LDDH probability decreases substantially over some areas of Northwest China and central-eastern China. Overall, a higher frequency of LDDH events is detected for the present climate compared to the reference period, with a clear shift of the whole distribution towards lower return periods (Fig. 4d).

To investigate the causes of variations in grid-scale LDDH events, we first show the temporal changes of annual maximum DUR and MAG during summer of 1961-2014 (Fig. 5). Consistent with the southern flood and northern drought pattern of summer precipitation over eastern China (Fig. 5c), maximum DUR has increased in Northeast China and significantly decreased over the Yangtze River valley (Fig. 5a). The changes in drought duration have been explained to be attributed to anthropogenic climate change (Sarhadi et al., 2018; Zhang et al., 2017) and natural variability such as the Pacific Decadal Oscillation (McCabe et al., 2004; Zhang and Zhou, 2015). A substantial increase in precipitation is associated with a pronounced decrease in dry spell duration over western Northwest China. This agrees with Shi et al. (2007), who reported a climate shift from warm-dry to warm-wet in the mid-1980s over the arid region of Northwest China, possibly related to an increase in atmospheric water vapour and an accelerated water cycle in response to global warming. Although with only a small decrease or even notable increase in temperature during DH events, most parts of Northwest China are four times less likely to experience long-lasting DH events (Fig. 4c), indicating the dominant role of a wetting climate and hence a shortened duration of dry spells in this region. The MAG shows a spatially similar but stronger warming trend than the summer mean maximum temperature over most areas of China, particularly over parts of Northeast China and the eastern edge of the Tibetan Plateau, with an increasing rate exceeding 0.4 °C per decade (Fig. 5b). This suggests that dry spells over these regions have warmed by at least 2 °C during the past 54 years, which may increase the risk of agricultural drought. In comparison, although mostly nonsignificant at the point level, the MAG of DH events has decreased by 0.2 °C per decade in some regions of central-eastern China. The combined effect of the decreases in both DUR and MAG may partly explain the

decreased probability of LDDH events over central-eastern China (Fig. 4c). Existing studies suggested that such cooling and wetting trends in central China during summer could be linked to a weakening of the East Asian summer monsoon (Yu et al., 2004; Zhou et al., 2009), which is related to the variations in sea surface temperatures of the Indian Ocean and the equatorial eastern Pacific (Gong and Ho, 2002; Hu et al., 2003; Zhao et al., 2010). Also, anthropogenic aerosols have been suggested to contribute to the observed changes in summer precipitation and temperature over eastern China (Dong et al., 2019; Li et al., 2007; Ye et al., 2013).

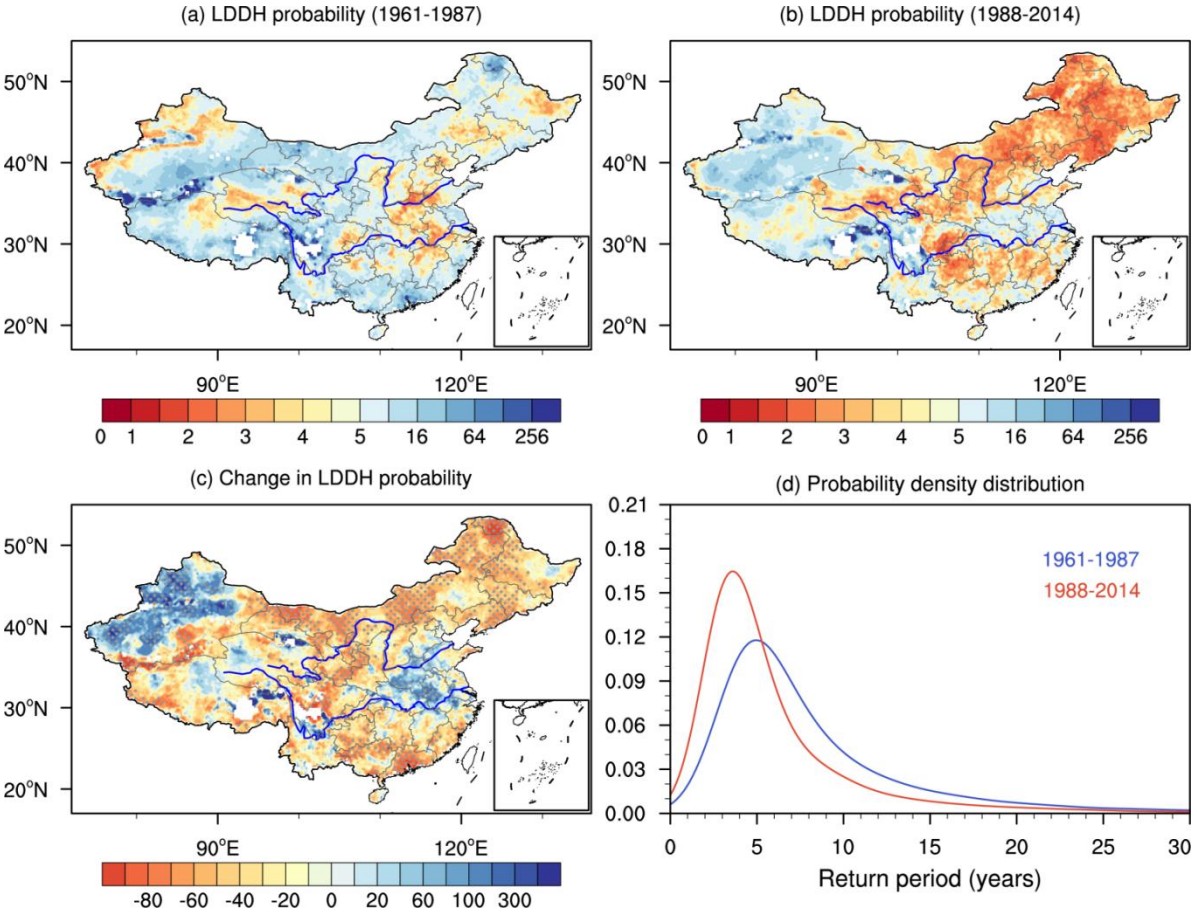

**Figure 4: Return periods (years) of compound long-duration dry and hot events for (a) 1961-1987 and (b) 1988-2014. (c) Percentage changes (%) across China for 1988-2014 relative to 1961-1987. Stippling indicates locations with significant changes at the 90% confidence interval. (d) Probability density functions based on all grid points for the two periods.**

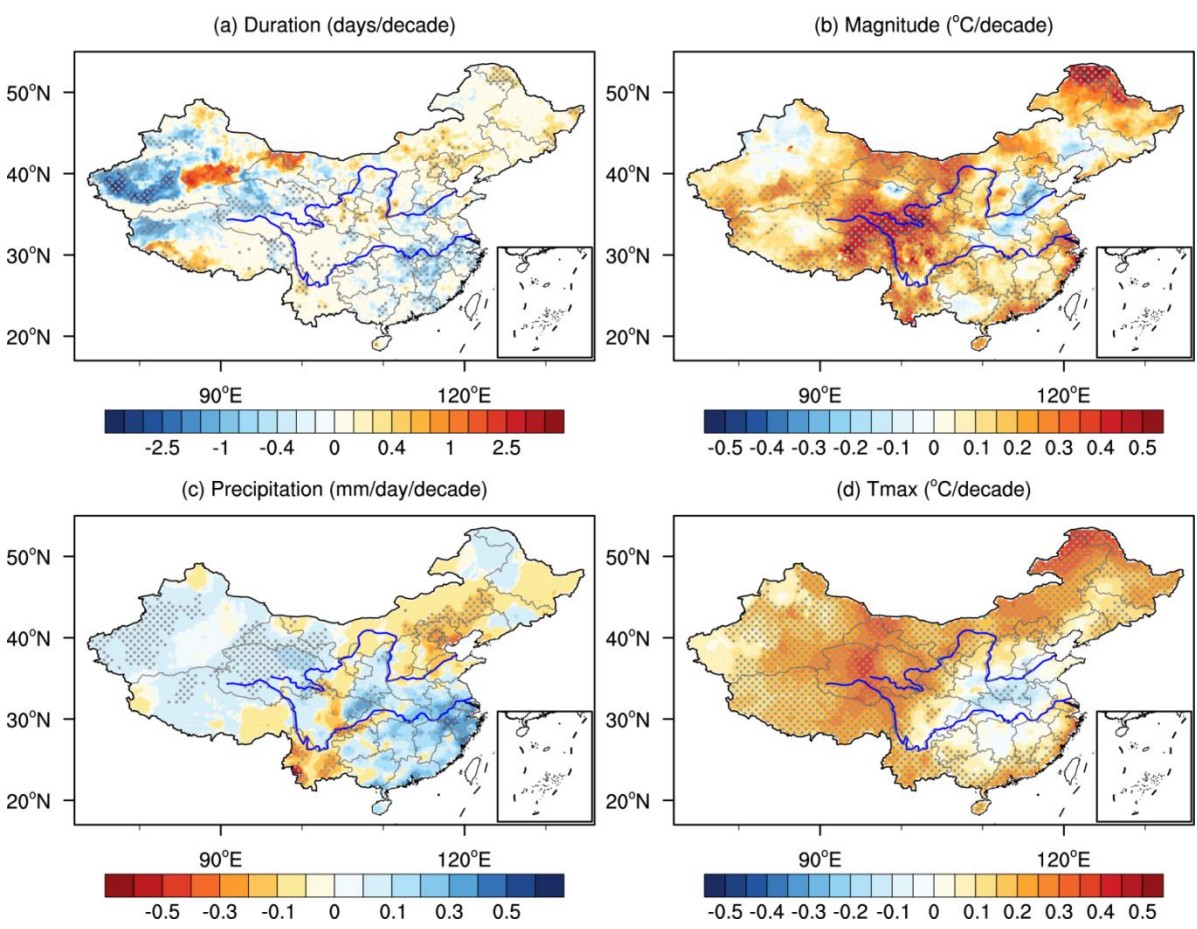

**Figure 5: Decadal trends for the summer maxima of (a) duration and (b) magnitude during 1961-2014. Trends for summer mean (c) precipitation and (d) Tmax. Stippling indicates significance at the 90% confidence level.**

In addition to changes in DUR and MAG, variations in the dependence between DUR and MAG may also influence LDDH risk. Following the method developed by Bevacqua et al. (2019), we quantitatively estimate the contributions from these potential drivers to the observed probability changes in persistent DH events (Fig. 6). Generally, changes in the MAG appear to be the primary driver of changes in LDDH probability. Increased MAG increases the concurrence frequency across most parts of China, with a relative contribution exceeding 67% ($\triangle$ RP = -40%) over regions stretching from Northeast to Southwest China, southern Tibet and southeastern China (Fig. 6b). A large contribution to changes in LDDH probability can be attributed to an increased dependence between DUR and MAG in some of these areas. However, over regions with a decreased frequency of LDDH events, the dominant drivers are different. For example, the DUR variability accounts for most of the variability of LDDH events in Northwest China, followed by the influence of MAG. Whereas the influence of decreasing temperatures prevails in most parts of central-eastern China, the decrease in LDDH events along the lower reaches of the Yangtze River is mainly related to a decrease in dependence and shortened dry spells. These decreases in the dependence between DUR and MAG are overall consistent with the findings of He et al. (2015) and Hao et al. (2019), who

demonstrated a weakened negative relationship between precipitation and temperature during summer in part of eastern China. Generally, large increases in MAG dominate the increased probability of LDDH events, highlighting the predominantly thermodynamic response of LDDH events to global warming.

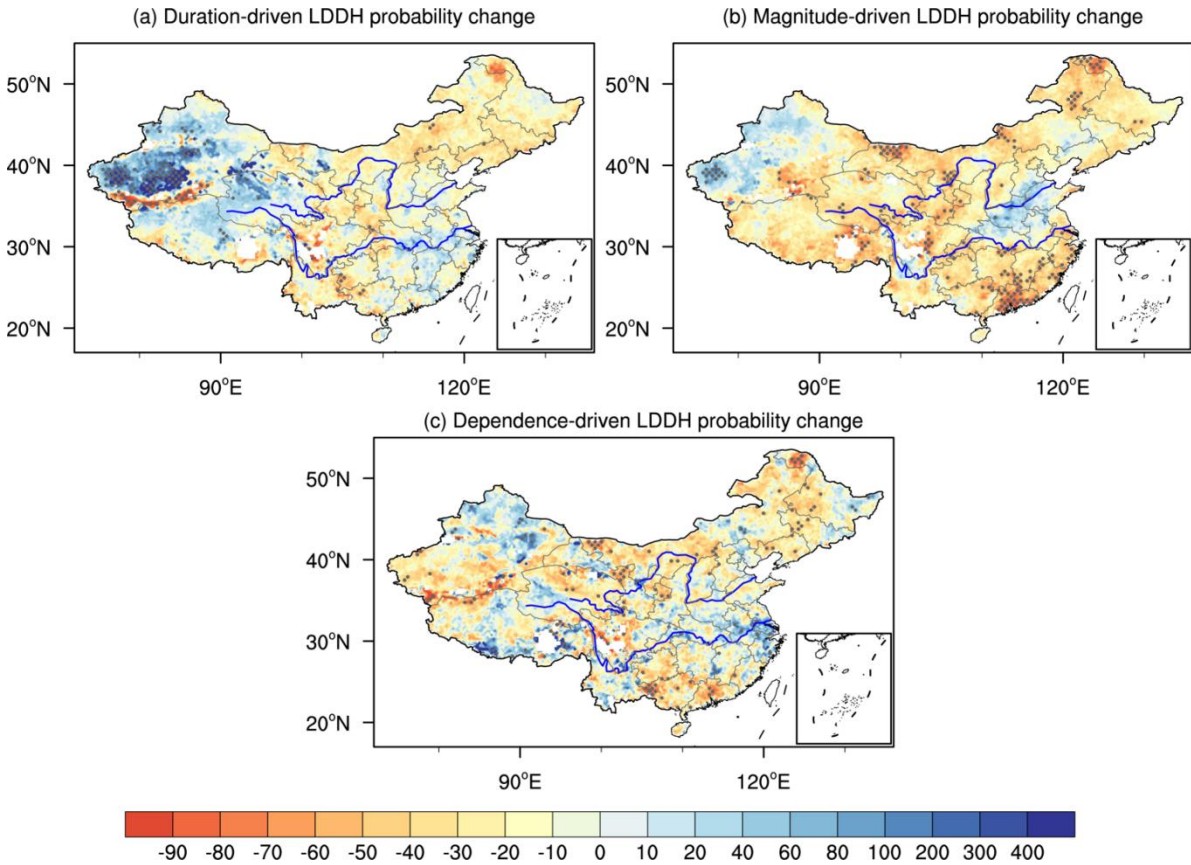

Figure 6: Changes (%) in return periods of compound long-duration dry and hot events between 1988-2014 and 1961-1987 due to changes in the marginal distribution of (a) duration, (b) magnitude, and (3) dependence between duration and magnitude. Statistically significant changes are shown by stippling.

## 3.2 Spatiotemporal compound long-duration dry and hot events

### 3.2.1 Climatological characteristics

In addition to the grid-based results, an event-based analysis is conducted to further understand the spatiotemporal evolution of compound extremes. In total, 146 spatiotemporal LDDH events are identified during 1961-2014. There is a significant positive correlation between the mean area and the mean duration of the spatiotemporal LDDH events (Fig. 7). This indicates that the SLDDH events with a longer duration are likely to have a larger spatial extent. The boxplot shows that the mean duration of SLDDH events is generally 6-11 days (interquartile range), with a median of about nine days, while few extreme events can last for more than 30 days. Note that the mean duration of the spatiotemporal events refers to the average duration of all cells during the event. When looking at the number of days between the onset and the end day, the duration of

the SLDDH events mainly ranges between 7 and 14 days, with a median of 10 days (not shown). The mean contiguous area impacted by SLDDH events varies between 200,000 and 330,000 km$^2$.

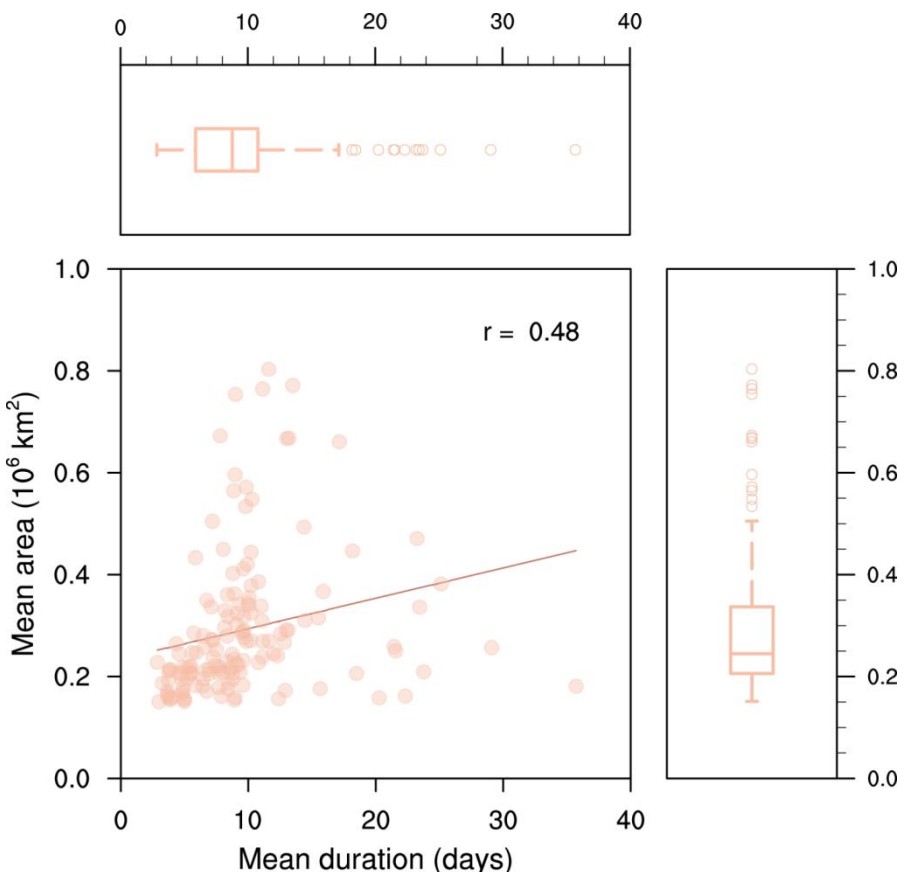

Figure 7: Scatter plot of mean duration and mean area for each spatiotemporal compound long-duration dry and hot event during 1961-2014. The red line is a least-squares line of best fit. Spearman correlation coefficient (r = 0.48, p < 0.05) is shown in the figure. The upper and right panels show the boxplot of mean duration and mean area, respectively. The box represents the interquartile range with the horizontal line indicating the median, whiskers expand from the minimum to the maximum and outliers are shown as circles.

Variations of the annual characteristics of the SLDDH events are presented in Fig. 8. Both the event number and mean area of SLDDH display a significant increasing trend during the past 54 years. The annual maximum value of mean duration slightly increases by 0.4 days per decade. Spatiotemporal LDDH events occur on average 2.7 times per year, increasing about 0.37 times per decade (Fig. 8a). The highest occurrence frequency of SLDDH is of 11 times and occurred in 2011. The severe 2011 hot and dry event over Southwest China was record-breaking and resulted in damaging impacts on crop production (Lin et al., 2015; Lu et al., 2014; Sun et al., 2012). The contiguous area affected by SLDDH events has expanded significantly with a rate of 40,000 km$^2$ per decade, according to the trend in annual maximum values (Fig. 8c). The largest spatial extension of SLDDH events is observed in the late 1990s to mid 2000s. The SLDDH events in 1970s and since 1990s have larger magnitudes. Overall, the spatiotemporal compound long-duration dry and hot events are becoming more frequent

and impacting larger areas. The increased contiguous areas affected by SLDDH could cause dramatic losses of agricultural production and populations in highly populated and agricultural regions (He et al., 2022; Zscheischler and Fischer, 2020), such as eastern China. A potential consequence of this fast-increased frequency and spatial expansion of persistent DH events is a growing threat to food production and electricity supply (Kim et al., 2022). In addition, more frequent and widespread large-scale SLDDH events may aggravate the risk of tree mortality and wildfires (Anderegg et al., 2013; Zscheischler et al., 2018), leading to an increase in the occurrence of sequential fire and dust extremes (Yu and Ginoux, 2022).

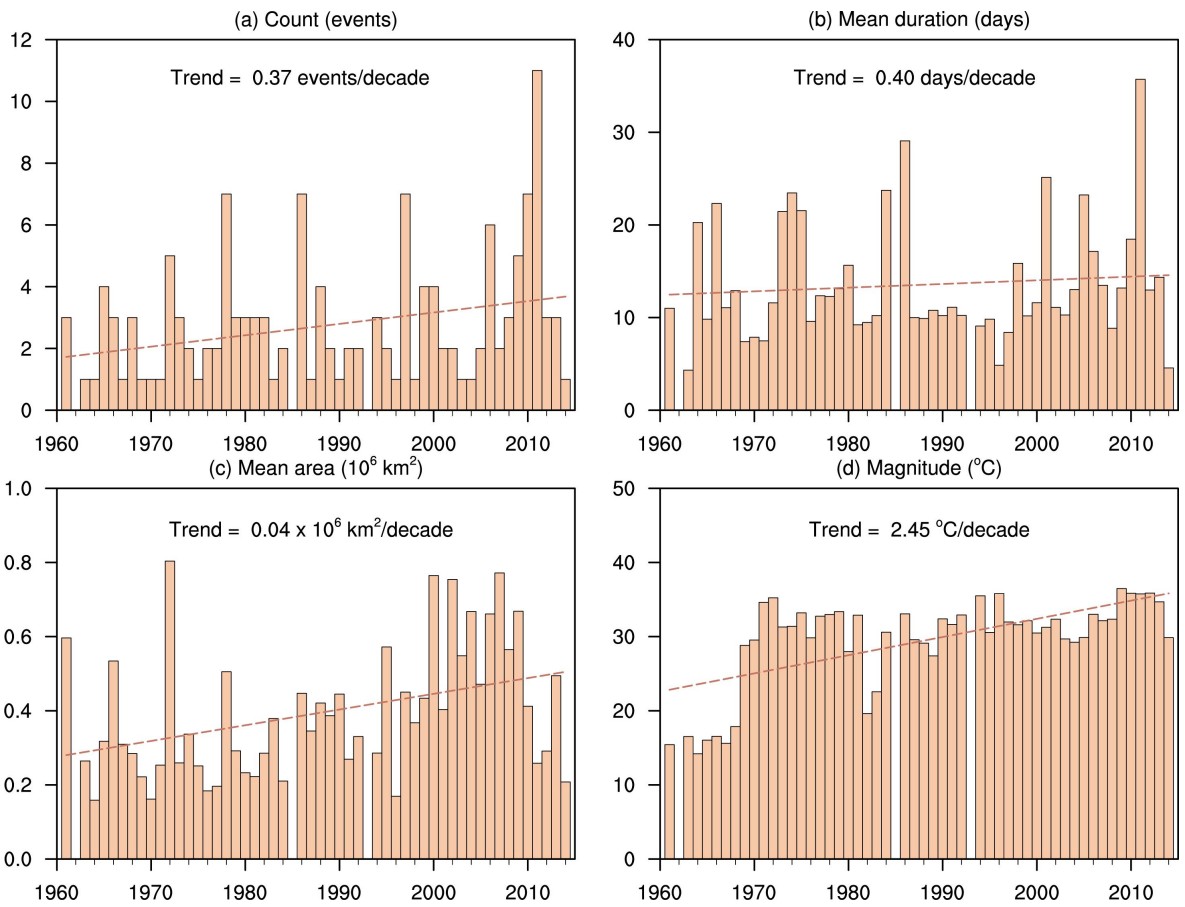

**Figure 8: Time series of the SLDDH (a) count, annual maximum value of (b) mean duration, (c) mean area, and (d) magnitude over 1961-2014. The dashed line shows the linear trend.**

### 3.2.2 Climatological characteristics

Using the hierarchical clustering algorithm (Sect. 2.3), nine typical SLDDH clusters are distinguished. The spatial distribution of these clusters is shown in Fig. 9. Shading indicates the percentage of SLDDH days affecting each grid point for each cluster. The nine clusters generally cover the whole of China, and are mainly located over the northern Tibetan Plateau (C1), North China (C2), Qinghai province (C3), Yunnan (C4), northern Xinjiang (C5), Northeast China (C6),

southern Xinjiang (C7), the middle and lower reaches of the YRB (C8), and Sichuan-Chongqing region (C9), respectively.

The C6 cluster centred over Northeast China occurs most frequently, including 31 spatiotemporal compound events, followed by the North China (C2, Fig. 9b) and Qinghai (C3, Fig. 9c) clusters, including 27 and 20 SLDDH events, respectively. Overall, the SLDDH events occur more frequently in northern China and less in southern China, which is consistent with the grid-based results.

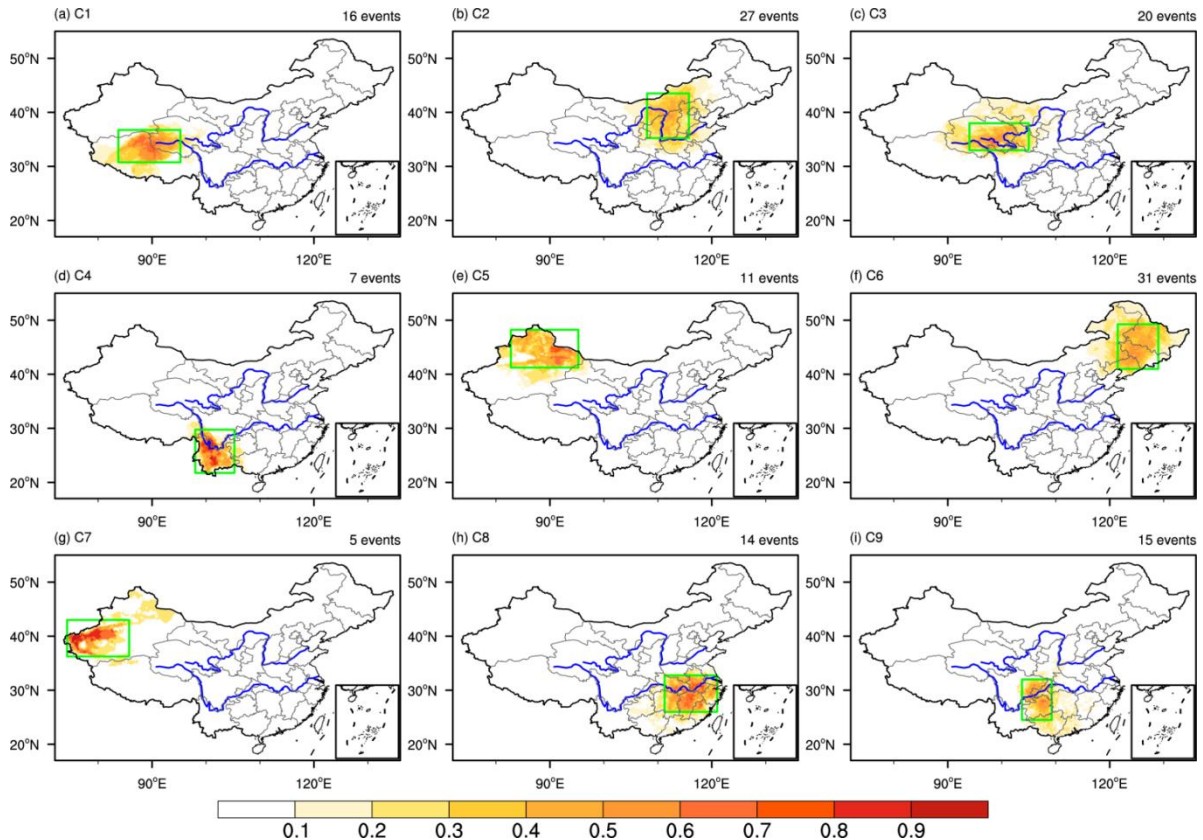

**Figure 9: The spatial pattern of nine SLDDH clusters in China. For each cluster, shading represents the percentage of SLDDH days affecting each grid point. The number of events in each cluster is listed in the upper-right corner of each panel.**

Although the Yunnan (C4) cluster only includes seven events, it has the highest temperature anomaly (> 4 °C) (Fig. 10d). Large high temperature anomalies are also apparent along the middle and lower reaches of the Yangtze River (i.e., C8 and C9, Fig. 10h-i). Consistent with the spatial pattern of temperature magnitude of grid-scale LDDH events (Fig. 2c), the

spatiotemporal events show higher magnitude in the northern Xinjiang (C5) and southern Xinjiang (C7) clusters (Fig. 11b). However, these two clusters have relatively weak temperature anomalies (Fig. 10e, g), which may be related to the high temperature climatology in those regions. The temperature anomaly of the C1 cluster over the northern Tibetan Plateau can reach up to about 3 °C, although the magnitude there is relatively low (approximately 15 °C) due to its high topography (Fig. 11b).

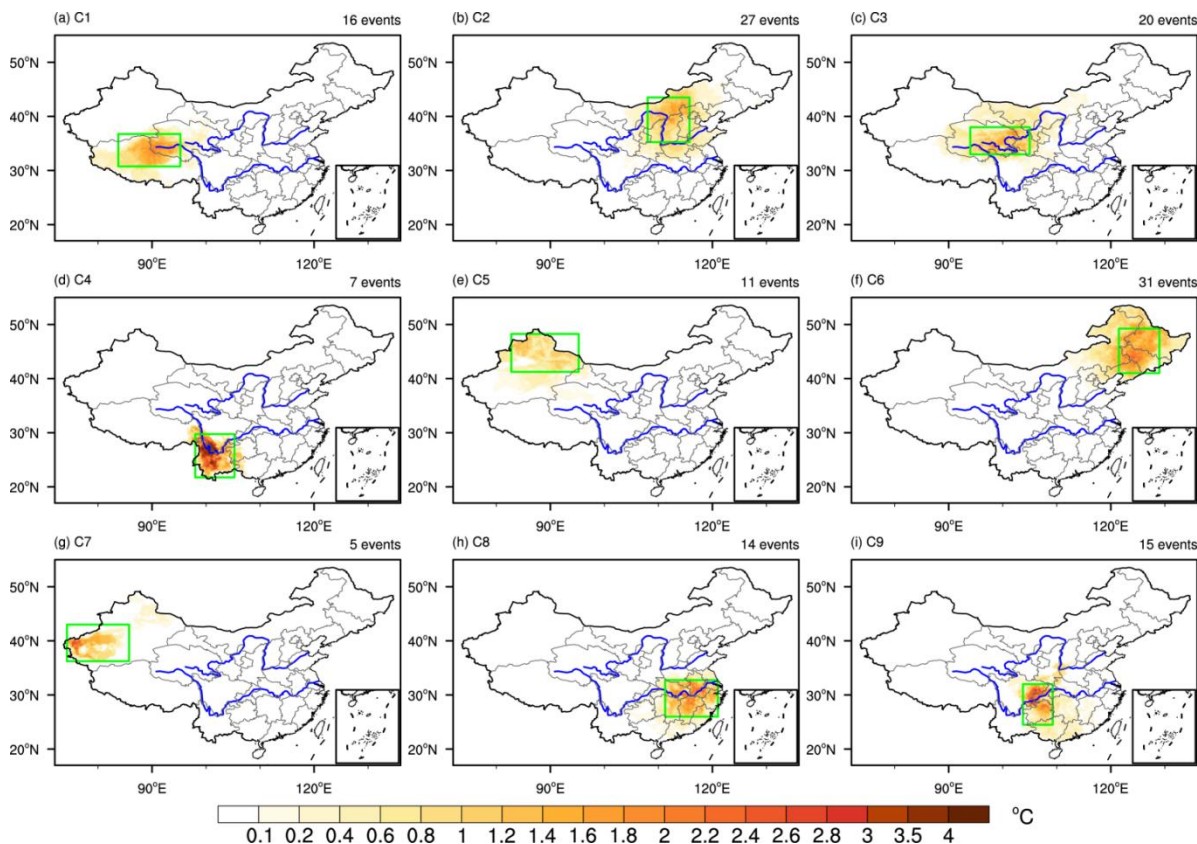

**Figure 10: Same as Figure 9, but for the average of daily Tmax anomaly (relative to the daily climatology) of all events in each cluster.**

The spatial extent (Fig. 11a) of SLDDH events in northern China is generally larger than in southern China. The contiguous affected area of SLDDH events over the northern Tibetan Plateau, in North China, Qinghai, and Northeast China (C1, C2,

C3 and C6) exceeds 300,000 km$^2$ per event. SLDDH events over the middle and lower reaches of the Yangtze River Basin (C8) also cover a large area, about 298,000 km$^2$. The Xinjiang region (C5 and C7) has the longest SLDDH events, with an average duration of up to 18 days, while Yunnan (C4) displays the shortest SLDDH events (about 5 days). Hotspots of frequency occurrence occur in northern regions such as Northeast China (C6), North China (C2) and Qinghai (C3). Large spatial variations are prevalent in the total days of SLDDH events (Fig. 11d). The highest total number of SLDDH days

exceeding 300 days are found in North China (C2) and Northeast China (C6), which may result from the higher occurrence frequency of SLDDH events in these areas (Fig. 11c). Relatively high values of SLDDH days reaching 269 days are also seen over the northern Xinjiang (C5), which may be attributed to the longer duration there. Conversely, only 42 days are affected by SLDDH events over Yunnan (C4).

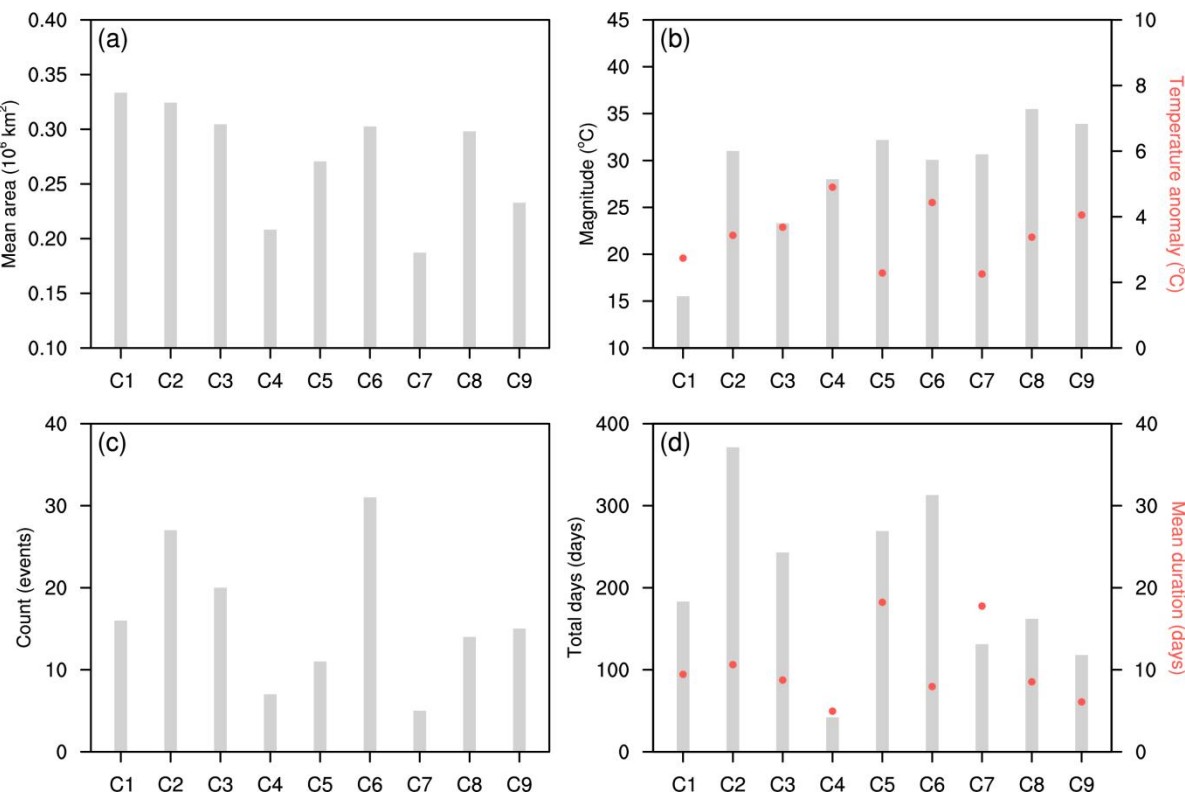

Figure 11: Characteristics of the nine SLDDH clusters during 1961-2014. The average (a) mean area, (b) magnitude (histogram) and temperature anomaly (dot), (c) count, and (d) total days (histogram) and mean duration (dot) of all events per cluster.

There is a clear seasonality of the SLDDH events, with most events (45.9%) begining in June (Fig. 12a). Note that the events are identified during summer from June to August. This might contribute to the low number of events in August. We test the sensitivity of our results to the cutoff by comparison with the results considering the events overlapping these months that end after this period. We find that the seasonality of SLDDH events are robust to the cutoff days of the analysis period (not shown). Whereas most of the events over northwestern regions (i.e., Xinjiang and Qinghai, C3, C5, and C7) and YRB (C8) generally start in July (Fig. 12b). The SLDDH events in the Sichuan-Chongqing region (C9) mainly occur from July to August, and peak in August. The onset timing of SLDDH is similar to that of grid-scale LDDH events (Fig. 2e), reflecting the influence of the summer monsoon and topography in different regions.

The occurrence of SLDDH events shows an obvious inter-decadal variability in the past 54 years (Fig. 12c). For the western region of China, the events over southern Xinjiang (C7) mainly occur in the 1970s and rarely happen after the 1990s, whereas 64% of events in northern Xinjiang (C5) occur from the 1960s to 1980s. Spatiotemporal LDDH events in the northern Tibetan Plateau (C1) are relatively more frequent in the 1980s and 1990s. The events over Yunnan (C4) occur much more frequently after 2010, which is consistent with the increase in the frequency of high temperature and droughts (Gao et al., 2021; Yang et al., 2019a). For eastern China, the frequency of SLDDH events has increased substantially after the 1990s,

especially in the Sichuan-Chongqing region (C9), where the SLDDH events rarely occur before the 1990s. For the middle and lower reaches of the Yangtze River (C8), SLDDH events are more frequent in the 1960s to 1980s and 2000s, but less frequent in the 1990s.

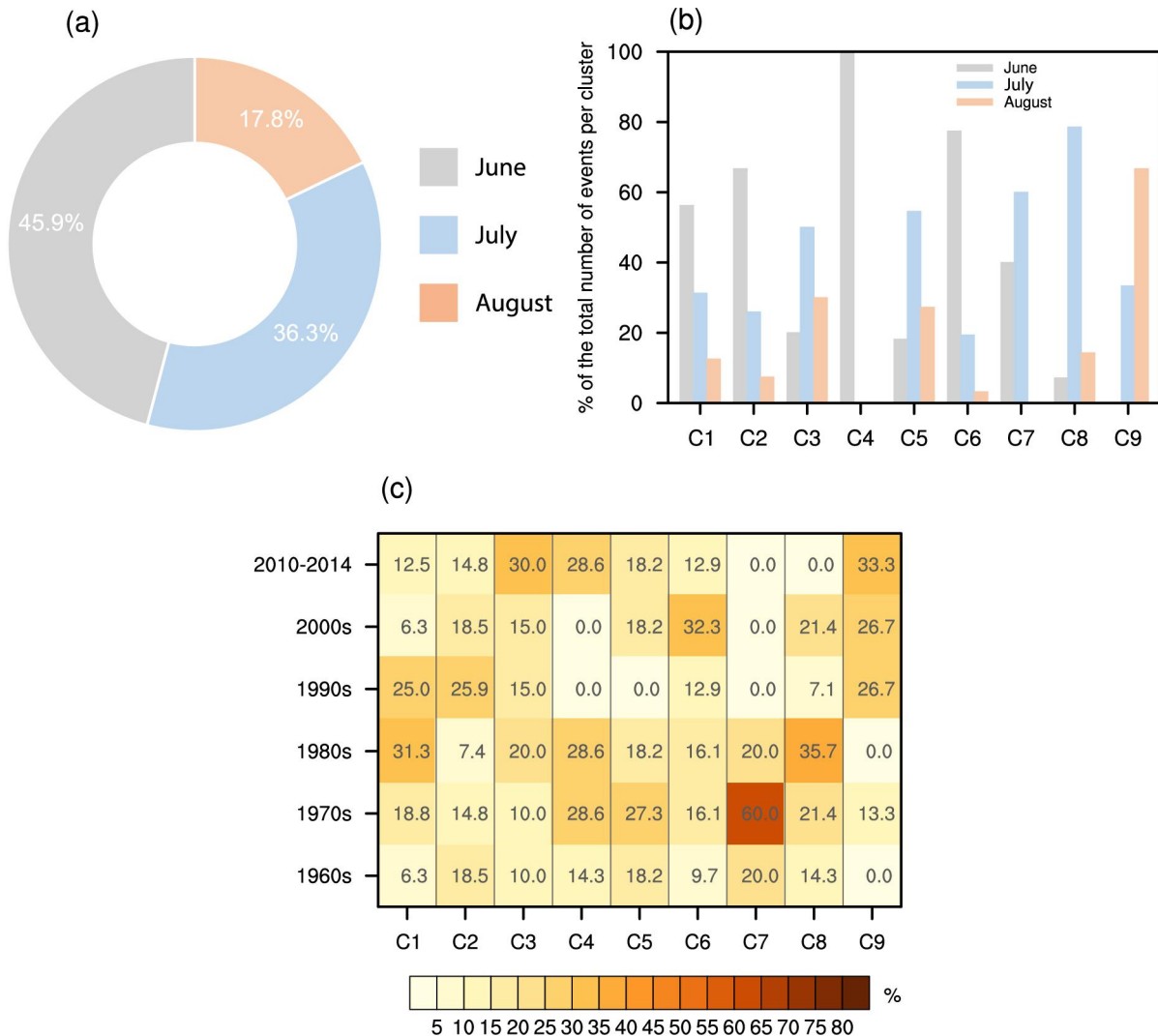

Figure 12: (a) The seasonality of the SLDDH events; the number shows the percentage of events starting in each month. (b) The percentage of events starting during each month for each cluster. (c) Percentage occurrence of SLDDH events for each cluster in different decades.

## 4 Summary and Conclusions

We investigate the spatiotemporal characteristics and occurrence probability of long-duration meteorological droughts that co-occur with extreme temperatures during summer in China. Unlike most studies that mainly focused on monthly or longer

timescale drought background, we identify dry spells at a daily timescale and focus on the persistence of co-occurring dry and hot conditions. Compound long-duration dry and hot events are investigated on both a grid and 3D event basis.

During 1961-2014, grid-scale LDDH events primarily occur in eastern China, especially in the northern part of the country. Hotspots include northeastern China and the middle and lower reaches of the Yangtze River Basin. Although Xinjiang is a region with few LDDH events, the events there are most persistent and accompanied by the highest temperature magnitudes. Notably, high temperature magnitudes are also apparent in central-eastern China. Averaged across the country, the frequency of the grid-scale LDDH events in China has increased significantly in the past 54 years. Increases in LDDH frequency are found across much of the country, most notably in Northeast China, Inner Mongolia, the Sichuan Basin, and China's southeast coast. However, the likelihood of LDDH events has decreased over most of Northwest and central-eastern China.

The increased likelihood of LDDH events is mainly associated with increasing temperatures. Similar results have been found in Europe (Manning et al., 2019) and the United States (Alizadeh et al., 2020), where global warming dominates the increase in the concurrent droughts and heatwaves in recent decades. An exception is found in western Northwest China, where shortened dry spells are the primary driver of the decreased frequency of LDDH events. Similarly, change in DUR contributes more than that of temperature along the lower reaches of the Yangtze River. Our quantitative analysis corroborates previous studies (Kong et al., 2020; Yu and Zhai, 2020), which highlighted the role of a reduction in drought occurrences in the decrease in DH events in the south-central region. Moreover, we find a predominant effect of decreases in the dependence on the reduction in DH probability in the southern central region. The underlying mechanisms for the changes in dependence structure (Hao et al., 2019), especially tail dependencies of compound extremes (Zscheischler et al., 2021), require further study.

A total of 146 spatiotemporal LDDH events are identified and grouped into nine clusters. The large-scale SLDDH events mainly occur in northern China, such as Northeast China, North China and Qinghai, where the events generally cover a larger area than other regions, with an average affected area greater than 300,000 km$^2$. Although the SLDDH events centred over Yunnan have the shortest duration, a relatively small affected area, and low frequency, the temperature anomaly is severe and can reach up to around 5 °C. The onset timing of SLDDH events has obvious seasonality. SLDDH events mostly begin in June, while the events in northwestern China and YRB generally start in July. There is a substantial increase in the occurrence frequency and annual maximum spatial extent of SLDDH events, with magnitudes of 0.37 times and 40,000 km$^2$ per decade, respectively. Both grid-scale and 3D analyses show a more frequent occurrence of compound long-duration dry and hot events under past warming conditions, indicating an increased probability of long-lasting compound events.

Our study provides a first characterization of where compound long-duration dry and hot events might occur in China, and how these events have changed. While most existing studies analyze DH events on a grid or station basis, we investigate the spatiotemporal variation of LDDH events from a 3D event-based perspective by considering connectivity in both space and time. Compound long-lasting dry and hot extremes occur more frequently with global warming, and quantitative detection and attribution analysis could help better understand the role of different anthropogenic forcings. It should be noted that our

results are based on gridded data, which may have uncertainties due to station density, especially over western China with relatively few stations. It would be interesting to compare the results with those of station observations. Although we find a dominant role of rising temperatures for the increased probability of LDDH events, improved understanding of the physical drivers (e.g., atmospheric conditions and land-atmosphere feedbacks) of the identified changes is essential for event prediction and risk management of such persistent compound events. Further studies could extend the approach taken here to identify the characteristics of compound long-duration dry and hot events in future climates and their impacts considering regional vulnerability and exposure.

**Data availability**

The observational dataset is available at http://data.cma.cn.

**Author contributions**

YY, DM, AO and JT conceived the study. YY analyzed the data and wrote the manuscript draft. DM, AO and JT provided comments and revised the manuscript. All authors discussed the results and contributed to the final manuscript.

**Competing interests**

The authors declare that they have no conflict of interest.

**Acknowledgements**

This work is supported by the National Key Research and Development Program of China (2018YFA0606003) and the Jiangsu Collaborative Innovation Center for Climate Change. Yi Yang is supported by the China Scholarship Council and the program A for Outstanding PhD candidate of Nanjing University.

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
