# Peer review of "Increased spatial extent and likelihood of compound long-duration dry and hot events in China, 1961-2014"

_Natural Hazards and Earth System Sciences, 2022_

## Author Comment (AC1)

**Reply to Reviewer #1 comments**

The paper proposes a very interesting analysis on the occurrence of combined hot and dry events in China, which represent a significant natural hazard. To do so, the authors use daily maximum temperature and precipitation data obtained from the CN05.1 dataset over the period 1961 to 2014.

In addition to the identification of hotspots for the country, the authors propose an analysis of the trends with which these events occurred and a further analysis of two sub-periods (1961-1987 and 1988-2014) to highlight traces of climate change.

I find the work to be well organised, well written and of high quality. In the following, I present some general recommendations that I think would be useful to improve the manuscript. In particular, I would extend the discussion not so much with regard to the results obtained (already very good), but by improving the narrative concerning the relationship between the extreme event studied <-> natural hazard (and the main purpose of the review).

Response: We gratefully acknowledge the reviewer's comments and the time spent to review the manuscript. The helpful and constructive comments have contributed to improve the paper. We have paid particular attention to improving the discussion on the potential impacts of such extreme events. For example, the substantial increase in the frequency of LDDH events identified in our results indicates increasing climate-related risks in China. The increased spatial extent of large-scale SLDDH events suggests increased population exposure and energy demand. Given the rapid population growth and land-use change, an increasing exposure and vulnerability to these events is expected. These could cause adverse impacts on agricultural, energy production and human health. The detailed responses can be found below.

- Section 3: change the title from 'Results' to 'Results and Discussion'.

Response: Thank you, the title has been changed to 'Results and Discussion'.

- In the introduction, I suggest adding at least one sentence on technologies and methods that offer large-scale climate data with sufficient resolution to obtain

practical information. Mention, for example, satellite products and large weather station networks such as the one used by the authors.

Response: We agree that greater detail on large-scale climate data would be helpful here and have added the following interpretations in bold to the manuscript (Line 52-53):

"… An event-based identification of spatiotemporal LDDH (SLDDH) could facilitate tracking the daily spatiotemporal dynamics of SLDDH and understanding the associated physical drivers. **The detection of SLDDH needs data with sufficient resolution that provide large-scale climate information, such as satellite measurements and large weather station networks.**"

• Although it is mentioned that such combined events can be both an environmental and human problem, I would strongly suggest strengthening the discussion on their role as a natural hazard. In other words, I would enrich the (well-done) climate analysis with indications more closely related to the concept of hazard (for example, more space for potential impacts). I believe that in this way the work can be further aligned with the aims of the journal.

Response: Thank you very much for this suggestion. We have provided more discussion on the potential impacts of compound dry and hot events from a natural hazard perspective. These changes can be found in the 'Results and Discussion' section as follows.

"... Overall, the spatiotemporal compound long-duration dry and hot events are becoming more frequent and impacting larger areas. **The increased contiguous areas affected by SLDDH could cause dramatic losses of agricultural production and populations in highly populated and agricultural regions (He et al., 2022; Zscheischler and Fischer, 2020), such as eastern China. A potential consequence of this fast-increased frequency and spatial expansion of persistent DH events is a growing threat to food production and electricity supply (Kim et al., 2022). In addition, more frequent and widespread large-scale SLDDH events may aggravate the risk of tree mortality and wildfires (Anderegg et al., 2013;**

**Zscheischler et al., 2018), leading to an increase in the occurrence of sequential fire and dust extremes (Yu and Ginoux, 2022).**"

**References**

Anderegg, W. R., Kane, J. M., and Anderegg, L. D.: Consequences of widespread tree mortality triggered by drought and temperature stress, Nat. Clim. Change, 3(1), 30-36, https://doi.org/10.1038/nclimate1635, 2013.

He, Y., Fang, J., Xu, W., and Shi, P.: Substantial increase of compound droughts and heatwaves in wheat growing seasons worldwide, Int. J. Climatol., 42(10), 5038-5054, https://doi.org/10.1002/joc.7518, 2022.

Kim, Y., Choi, Y., and Min, S. K.: Future changes in heat wave characteristics and their impacts on the electricity demand in South Korea, Weather and Climate Extremes, 37, 100485, https://doi.org/10.1016/j.wace.2022.100485, 2022.

Yu, Y., and Ginoux, P.: Enhanced dust emission following large wildfires due to vegetation disturbance, Nat. Geosci., 15(11), 878-884, https://doi.org/10.1038/s41561-022-01046-6, 2022.

Zscheischler, J., and Fischer, E. M.: The record-breaking compound hot and dry 2018 growing season in Germany, Weather and climate extremes, 29, 100270, https://doi.org/10.1016/j.wace.2020.100270, 2020.

Zscheischler, J., Westra, S., Van Den Hurk, B. J. J. M., Seneviratne, S. I., Ward, P. J., Pitman, A., Aghakouchak, A., Bresch, D. N., Leonard, M., Wahl, T., and Zhang, X.: Future climate risk from compound events, Nat. Clim. Change, 8, 469–477, https://doi.org/10.1038/s41558-018-0156-3, 2018.

- In my opinion, the most interesting part of the work is the fact that such extreme events are tending to increase in frequency with the passage of time. This fact of climate change should find more space in the conclusions.

Response: We have highlighted the observed increase in frequency of such compound extremes in the 'Summary and Conclusions' section as follows.

**"Averaged across the country, the frequency of the grid-scale LDDH events in China has increased significantly in the past 54 years. Increases in LDDH frequency are found across much of the country, most notably in Northeast China, Inner Mongolia, the Sichuan Basin, and China's southeast coast."**

**"There is a substantial increase in the occurrence frequency and annual maximum spatial extent of SLDDH events, with magnitudes of 0.37 times and 40,000 km² per decade, respectively. Both grid-scale and 3D analyses show a more frequent occurrence of compound long-duration dry and hot events under**

**past warming conditions, indicating an increased probability of long-lasting compound events."**

- Also in the conclusions, I would dedicate a few more sentences on possible limitations of the proposed approach.

Response: Thank you. More details on the potential limitations in the context of our research have been provided in the manuscript. We note some shortcomings in our analysis, mainly driven by data limitations. The gridded dataset may introduce biases due to spatial interpolation of station observations, especially over regions with sparse stations. Moreover, although we find a dominant role of rising temperatures for the increased probability of LDDH events, improved understanding of the physical drivers (e.g., atmospheric conditions and land-atmosphere feedbacks) of the identified changes is essential for risk management of such persistent compound events. To better understand impacts of such compound events, it would be valuable to link the identified changes in climate extremes to socioeconomic indicators of exposure, such as the affected population. Changes made in text include the following ones.

"... It should be noted that our results are based on gridded data, which may have uncertainties due to station density, **especially over western China with relatively few stations**. It would be interesting to compare the results with those of station observations. **Although we find a dominant role of rising temperatures for the increased probability of LDDH events, improved understanding of the physical drivers (e.g., atmospheric conditions and land-atmosphere feedbacks) of the identified changes is essential for event prediction and risk management of such persistent compound events. Further studies could extend the approach taken here to identify the characteristics of compound long-duration dry and hot events in future climates and their impacts considering regional vulnerability and exposure.**"

Again, we sincerely appreciate the opportunity to revise our work for consideration for publication in Natural Hazards and Earth System Sciences. Thank you once again

for your valuable comments and suggestions. We sincerely hope that our revision will meet with your approval.

---

## Author Comment (AC2)

**Reply to Reviewer #2 comments**

This article assesses long duration dry and hot events over China. The analysis is done locally at each grid but also includes 3D analysis of the events taking their duration, temperature magnitude and contiguous area into account. The paper presents the climatology of each characteristic and assesses changes in the characteristics over the observation period. The paper is well written and the results are nicely presented. I can recommend this paper for publication after some minor revisions that I have outlined below.

Response: We thank the reviewer for the time in reviewing this manuscript. The comments are quite valuable and helpful for revising and improving our paper. We have clarified some parts of the manuscript to make it clear and concise.

The authors point out that the gridded dataset is based on a number of underlying observation stations and briefly mention at the end of the manuscript that a comparison with station data would be interesting to assess in further studies. However, I think it is important to show or discuss the spatial distribution and density of these stations. This would highlight if some regions are more observed than others and whether we should trust the changes found in regions with low observation densities. For example, in Figure 12b, is it surprising that C7 has no events for three decades (from 1990 onwards)? How trustworthy is the data in this region?

Response: Thank you for pointing this out. We agree that the spatial distribution and density of the stations may play a role in the uncertainty of gridded datasets. The gridded observation dataset we used here is based on observations from over 2,400 meteorological stations in China. Compared to the high density of stations over eastern China, the stations over western China are relatively sparse, especially over the northern part of the Tibetan Plateau and Taklimakan desert in southern Xinjiang (the location of the C7 cluster), which leads to a great uncertainty in these regions.

We have added the following text to the manuscript.

"**Note that the station density in western China is lower than in eastern China, leading to a great uncertainty in this region, with the largest uncertainty over the**

**northern part of the Tibetan Plateau and Taklimakan desert in southern Xinjiang (Peng and Zhou, 2017; Wu and Gao, 2013).**"

"... It should be noted that our results are based on gridded data, which may have uncertainties due to station density**, especially over western China with relatively few stations**."

**References**

Peng, D., and Zhou, T.: Why was the arid and semiarid northwest China getting wetter in the recent decades?, J. Geophys. Res.-Atmos., 122(17), 9060-9075, https://doi.org/10.1002/2016JD026424, 2017.

Wu, J., and Gao, X. J.: A gridded daily observation dataset over China region and comparison with the other datasets (in Chinese), Chinese J. Geophys., 56(4), 1102-1111, https://doi.org/10.6038/cjg20130406, 2013.

P3 L71: I suggest replacing 'meteorological drought' with a 'dry spell'. The two terms seem to be used interchangeably throughout which might become confusing for readers as 'meteorological drought' has been defined in many different ways in the literature (e.g. some define meteorological drought with SPI), while a dry spell definition is more precise and will not become confused with different definitions.

Response: Thanks for this suggestion. We concur with the reviewer and have replaced 'meteorological drought' with 'dry spell' in Line 75 to make it precise.

P3 L89: The metric 'Count' can be difficult to interpret and I don't think it's the best metric to use when assessing events defined by their duration. For example, each summer has 92 days, if you have 1 event that lasts 80 days and rain on the remaining 12 days, you have 1 event. Equally, you may also have a wet summer with one event lasting 14 days, or multiple short duration events that leads to a high count. This is an issue in interpreting changes in the metric. For instance, a reduction in the number of events could mean they are less frequent or that are more persistent. Perhaps an explanation of this would help though I see that the analysis of event counts is supplemented with changes in the number of LDDH days (Figure 3), which is helpful.

Response: We agree with the reviewer that caution must be taken when assessing events defined by their duration. Therefore, in addition to the number of such persistent extreme events, other metrics such as the number of total LDDH days and duration are also considered in our analysis.

P5 L127-128: Perhaps you could be more precise here as well as give an example of what you mean by a dominant spatial pattern. Is it finding clusters of similar events in different regions within the analysis area or the shape of the patterns (etc.)? Also, please provide the motivation behind this cluster analysis and further discussion of the relevance of the results obtained from this.

Response: Thank you for this comment. The motivation for the cluster analysis is to distinguish clusters of these events following their geographical location. These spatial clusters allow us to investigate the characteristics of the large-scale SLDDH events in different regions. Our results show that SLDDH events in China exhibit distinct characteristics in different geographical locations. For example, SLDDH events in northern China are generally more frequent and have a greater spatial extent than those in the south. The mechanisms/processes for their occurrence may be regionally dependent. It is of great interest to extend the current analysis to examine the physical mechanisms associated with the different clusters, thus supporting the adaptive management of such extreme events.

We have clarified the clustering approach in more detail in section 2.3 as follows.

**"Here we apply a hierarchical clustering algorithm (Rokach and Maimon, 2005) to identify clusters of SLDDH events over China following their geographical location and analyze their associated characteristics."**

Section 3.2.1: This section could be improved, particularly the description of the metrics they assess and the justification for the metrics assessed in Figure 8. It's not instantly clear what exactly the authors are assessing here. My understanding is that they assess trends in the mean annual characteristics of events. This is not so informative without an assessment of the variability within seasons (e.g. larger/longer impactful events may be averaged out by smaller/shorter, and less impactful, events). Perhaps an assessment of the trend of the maximum event each year would be more informative. For instance, one could estimate a conditional trend for the area associated with the annual maximum duration event, or vice versa.

Response: Thank you for pointing this out. In previous submission, when assessing

trends in different characteristics (i.e., mean duration, mean area, and magnitude) of SLDDH events, the annual maximum is considered for mean duration, whereas the annual mean values of all events is used for spatial extent and magnitude.

We agree with the reviewer that trends calculated by considering all events may be influenced by the variability of events in each year. As suggested, we have updated Fig. 8 by considering trend of the annual maximum, the text and figure caption have been rephrased accordingly.

[Figure]

**Figure 8: Time series of the SLDDH (a) count, annual maximum value of (b) mean duration, (c) mean area, and (d) magnitude over 1961-2014. The dashed line shows the linear trend.**

Figure 7: It would be interesting to included magnitude in this figure also. Do larger area events have higher temperatures?

Response: Thank you for this comment. As shown in Fig. A1, the magnitude and spatial extent are not strongly correlated with each other. The events with a large areal extent can have moderate temperatures.

[Figure]

**Figure A1:** Scatter plot of magnitude and mean area for each regional compound long-duration dry and hot event during 1961-2014. The red line is a least squares line of best fit. Spearman correlation coefficient (r = -0.04, p > 0.05) is shown in the figure. The upper and right panels show the boxplot of magnitude and mean area, respectively.

P14 L294-5: 'The annual maximum mean duration' is confusing. I'm not sure what this means exactly, it should be clarified.

Response: The SLDDH events are characterized by their mean impacted area, mean duration and temperature magnitude. For each SLDDH event, mean duration is defined as the average duration of all cells contributing to the 3D event. In Figure 8, the annual maximum of this metric is considered. This sentence has been rephrased as "The annual maximum value of mean duration" in the revised manuscript.

Figure 12a: For events starting in August, do they automatically end at the end of August or are days in September counted also if the event persists past the end of August? If not, this might contribute to the low number of events in August.

Response: Thank you for pointing this out. Here we only consider the days during

summer from June to August (92 days in each summer). We agree with the reviewer that the cutoff at the 31st of August might have an impact on event detection, contributing to the low number of events in August. To test the sensitivity of cutoff on seasonality, two cases are considered. Case 1 only considers the days during summer of JJA, while Case 2 also includes the events overlapping these months that begin before or end after this period. Thus, the events that start in August and extend beyond this month are also counted in Case 2.

Generally, the number of events begin in August is low in both cases (Table A1), although 20% of events begin in May with a mean beginning date of May 24th (all of them ends in June) in Case 2. When considering the seasonality for different clusters, the results of the two cases are similar, so this choice does not affect the overall results of the study (Figure A1). An exception is found for C5 over Northern Xinjiang, where most events begin in August (July) in Case 2 (1), which corresponding to a total of 6 events in both cases.

**Table A1.** The percentage of events starting in each month (%).

| Month | 4 | 5 | 6 | 7 | 8 |
|---|---|---|---|---|---|
| Case 1 | 0 | 0 | 45.9 | 36.3 | 17.8 |
| Case 2 | 1.3 | 20 | 27.3 | 30.7 | 20.7 |

[Figure]

**Figure A2:** The percentage of events starting during June, July, and August for each cluster in (a) Case 1 and (b) Case 2. Case 1 only considers the days during summer of JJA, while Case 2 also includes the events overlapping these months that begin before or end after this period.

This has been discussed in the manuscript as follows.

"... There is a clear seasonality of the SLDDH events, with most events (45.9%) begining in June (Fig. 12a). **Note that the events are identified during summer from June to August. This might contribute to the low number of events in August. We test the sensitivity of our results to the cutoff by comparison with the results considering the events overlapping these months that end after this period. We find that the seasonality of SLDDH events are robust to the cutoff days of the analysis period (not shown).** "

Finally, we appreciate all of your insightful comments. Thank you for taking the time to help us improve the paper. We are hopeful that our revised version rises to your expectations.